# Gas Sensing and Half-Metallic Materials Design Using Metal Embedded into S Vacancies in WS_2_ Monolayers: Adsorption of NO, CO, and O_2_ Molecules

**DOI:** 10.3390/ijms242015079

**Published:** 2023-10-11

**Authors:** Eduardo Rangel-Cortes, José Pablo Garcia-Islas, Josue Gutierrez-Rodriguez, Saul Montes de Oca, José Andres Garcia-Gonzalez, José Manuel Nieto-Jalil, Alan Miralrio

**Affiliations:** Escuela de Ingeniería y Ciencias, Tecnologico de Monterrey, Ave. Eugenio Garza Sada 2501, Monterrey 64849, N.L., Mexico; a01039660@tec.mx (J.P.G.-I.); a01754207@tec.mx (J.G.-R.); saul.armeaga@tec.mx (S.M.d.O.); anteus79@tec.mx (J.A.G.-G.); jnietoj@tec.mx (J.M.N.-J.)

**Keywords:** half metallic behavior, tungsten disulfide, gas sensor, molecular adsorption, atmospheric pollutants

## Abstract

The adsorption of CO, NO, and O_2_ molecules onto Cu, Ag, and Au atoms placed in the S vacancies of a WS_2_ monolayer was elucidated within dispersion-corrected density functional theory. The binding energies computed for embedded defects into S vacancies were 2.99 (*Au_S_*), 2.44 (*Ag_S_*), 3.32 eV (*Cu_S_*), 3.23 (*Au_2S2_*), 2.55 (*Ag_2S2_*), and 3.48 eV/atom (*Cu_2S2_*), respectively. The calculated diffusion energy barriers from an S vacancy to a nearby site for Cu, Ag, and Au were 2.29, 2.18, and 2.16 eV, respectively. Thus, the substitutional atoms remained firmly fixed at temperatures above 700 K. Similarly, the adsorption energies showed that nitric oxide and carbon oxide molecules exhibited stronger chemisorption than O_2_ molecules on any of the metal atoms (Au, Cu, or Ag) placed in the S vacancies of the WS_2_ monolayer. Therefore, the adsorption of O_2_ did not compete with NO or CO adsorption and did not displace them. The density of states showed that a WS_2_ monolayer modified with a Cu, Au, or Ag atom could be used to design sensing devices, based on electronic or magnetic properties, for atmospheric pollutants. More interestingly, the adsorption of CO changed only the electronic properties of the MoS_2_-*Au_S_* monolayer, which could be used for sensing applications. In contrast, the O**_2_** molecule was chemisorbed more strongly than CO or NO on *Au_2S2_*, *Cu_2S2_*, or *Ag_2S2_* placed into di-S vacancies. Thus, if the experimental system is exposed to air, the low quantities of O_2_ molecules present should result in the oxidation of the metallic atoms. Furthermore, the O_2_ molecules adsorbed on WS_2_-*Au_2S2_* and WS_2_-*Cu_S_* introduced a half-metallic behavior, making the system suitable for applications in spintronics.

## 1. Introduction

Metal oxides, such as SnO_2_ and ZnO, have been employed and commercialized for several years to create highly effective gas sensors at low cost. In addition, it has been found that the reduction in the size of the crystal increases the sensitivity and performance of the sensors [1,2,3,4]. These chemiresistive sensors have been primarily used due to the ability of their wide-band gap to serve as a sensing medium [5]. However, such sensors often require high operating temperatures (200–500 °C), which would require high energy consumption, raising potential security concerns [6]. Since the appearance of this technology, the number of studies of gas sensing materials, particularly transition metal dichalcogenides which can operate at room temperature, has been increasing [7,8,9,10,11,12,13,14]. Our research group recently demonstrated that a molybdenum disulfide monolayer (ML) modified with group 11 dimers placed in S vacancies could be used to create novel materials which, based on the adsorption of nitric oxide, carbon monoxide, and molecular oxygen, would act as electronic or magnetic sensors [15]. Moreover, the presence of nitric oxide and oxygen molecules on MoS_2_-*Au_2S2_* and MoS_2_-*Cu_2S2_* leads to the transition to a half-metallic state, making these materials suitable for spintronics applications [15]. Charge transfer occurs from the monolayer to the adsorbed molecules, oxidizing atmospheric components and pollutants. This has been explained in terms of the lower ionization potentials of the substrates as well as the higher electronic affinities of the molecules. Thus, the substitutional atoms at S sites [16,17], the formation of vacancies [13,18,19,20], and the usage of several substrates [21] have served as the bases to enhance the chemical reactivity of monolayers. Cui et al. recently showed that the adsorption of CO, NH_3_, NO, and NO_2_ and the W and S vacancies can modulate the magnetic and electronic properties of a tungsten disulfide monolayer [13]. Moreover, research on the magnetic, electronic, and optical behaviors of a tungsten disulfide monolayer after metal adsorption was reported [12]. The observed adsorption energies suggest that several atoms anchored on the WS_2_ monolayers could be stable at room temperature. However, to date, study of the energetic stability of metal atoms adsorbed on WS2-ML via a reaction path, known as metallic diffusion, from the most stable adsorption site to neighboring sites, has not been carried out. Then, metal atoms could tend to coalesce into larger clusters. Moreover, WS_2_ has been utilized in the fabrication of chemiresistive sensors due to its demonstrated significant response to various target analytes, including toluene, acetone, ethanol, and water [14].

Compared to MoS_2_ ML, WS_2_ monolayers are cheaper to produce and are environmentally friendly [22,23]. Currently, WS_2_ monolayers are synthesized using the chemical vapor deposition method [24] and the sulfurization of ultrathin WO_3_ films [25]. The formation of vacancies has been used to chemically activate the monolayer [26,27,28]. Thus, the adsorption of O_2_ molecules, which is a very reactive species present in the air with a concentration of approximately 20%, or of small atmospheric pollutants such as NO and CO (secondary greenhouse gases) could take place by the greater chemical reactivity achieved by defects on WS_2_ monolayers. If the ML is exposed to O_2_ molecules, it could be a competitor for carbon monoxide and nitric oxide adsorption, or these species could even react with it, forming other compounds. Moreover, defective electronic states, introduced by the aforementioned defects on WS_2_, could lead to the development of materials that are highly sensitive to atmospheric pollutants in the gas phase [12]. In addition, the chemisorption of small molecules could modify the electronic structure of the monolayer [13].

Since the use of transition metal dichalcogenides (TMDCs) with conductance near the quantum limit has been proposed for electronic devices [8,29,30,31,32], the scientific community has been focused on the analysis of ultra-clean contacts [33], as well as surface defects [34]. Since one of the most challenging problems facing TMDC devices is the high contact resistance, several metals have been tested [34,35]. Matching alignments, as well as contact metals, could lead to chemisorption or physisorption. For instance, Wang and coworkers determined that Ag leads to Ohmic contacts with the MoS_2_ bulk and forms van der Waals interactions in the interface with MoS_2_ bilayers [36]. Additionally, Ohmic contacts have been reported for WS_2_/Zr_2_C and WS_2_/Hf_3_C_2_ systems [37]. Similarly, ab initio calculations were performed on WSe_2_ [35].

The changes introduced by sulfur vacancies on WS_2_ monolayers have been studied, even by experimental approaches. For instance, Schuler reported micrographs of CVD-grown monolayers of WS_2_ obtained using CO-tip non-contact atomic force microscopy and scanning tunneling microscopy/spectroscopy (STS). The results obtained by STS revealed a characteristic fingerprint caused by an extraordinarily strong spin-orbit coupling, confirmed by DFT and GW calculations [38]. Similar behaviors have been identified in the case of Mn-doped monolayers of WS_2_, in which impurity *d* states appeared within the bandgap. Magnetic properties were introduced as well [39].

Gas sensing using two-dimensional materials exploits charge transfer processes that modify the resistance after exposure to gas molecules [40,41]. For instance, in a previous report, density functional theory was used to investigate the adsorption of CO, NH_3_, NO, and NO_2_ molecules on a monolayer, bilayer, and heterobilayer of MoS_2_ and WS_2_ [9]. The calculated adsorption energies showed that the gas molecule was physisorbed in each case, with values ranging from 90 to 200 meV. According to the Arrhenius law, these energies suggest that the molecules could not be absorbed at room temperature. In addition, it has been reported that the adsorption of CO, NH_3_, NO, and NO_2_ molecules on defects can modulate the magnetic and electronic properties of WS_2_ [13]. However, the calculated adsorption energies suggested that the molecules could not be absorbed at room temperature, except for NO, which was adsorbed on *V_W_* and *V_S_* defects with 0.88 and 0.31 eV, respectively.

Thus, in the present study, dispersion-corrected density functional theory (DFT) was used to establish the adsorption behavior of molecules (AB = O_2_, CO, and NO) on coinage atoms M (M = group 11) placed into mono-(*V_S_*) and di-vacancies (*V_2S2_*) of S on WS_2_ MLs. For the first time, this report elucidates the electronic and magnetic properties of these systems as a means of determining the modulation of gas sensors by the adsorption of toxic gas molecules, providing a theoretical basis for the fabrication of WS_2_ gas sensors. The modulation of coinage metal adsorbed into mono-(*V_S_*) and di-vacancies (*V_2S2_*) of S on WS_2_ MLs has not yet been systematically investigated regarding electronic, magnetic, and optical behaviors. In addition, half-metallic systems, which would be suitable for use in spintronics and sensing devices, were obtained. Moreover, the present report discusses the results of the adsorption energies, Bader charges, and the total (DOS) and projected (PDOS) density of states used to describe molecule−sensor interactions.

## 2. Results and Discussion

The lattice parameter of WS_2_ was computed as 3.19 Å, which is consistent with the results of previous reports [41,42] (Figure 1a). A band gap of 1.85 eV was calculated for pristine WS_2_ (see Figure 1). The direct gap of the WS_2_ monolayer, i.e., about 1.9 eV, has also been experimentally confirmed by another group [25]. Likewise, coinage metal atoms *M_S_* placed into sulfur vacancies are located above the monolayer (Figure 1).

The calculated binding energies for metal atoms placed into S vacancies were as follows: 2.99 (*Au_S_*), 2.44 (*Ag_S_*), 3.32 (*Cu_S_*), 3.23 (*Au_2S2_*), 2.55 (*Ag_2S2_*), and 3.48 eV/atom (*Cu_2S2_*). The corresponding net charges were +0.15, +0.1 and +0.18 e for *Cu_S_*, *Au_S,_* and *Ag_S_*, respectively. These results evidence that substitutional atoms were attached to their respective vacancies. The strong bonding of Cu, Ag, or Au atoms with the S vacancy of the WS_2_ monolayer could prevent the diffusion problem by imposing a high barrier. To confirm this, the diffusion of Cu, Au, or Ag atoms from S vacancies to a nearby site (hexagon ring) was investigated, as shown in Figure 2.

The calculated diffusion energy barriers, obtained based on the nudged elastic band (NEB), from S vacancy as an initial state (IS) to the nearby hexagonal site as a final state (FS), for Cu, Ag, and Au were computed as 2.29, 2.18, and 2.16 eV, respectively. These values were large enough; the reaction energies were 1.38 (Cu), 1.71 (Ag), and 1.88 (Au) eV, indicating endothermic reactions (see Figure 2). These values were computed at the correspondent transition state (TS). On the other hand, the diffusion energy barriers from the nearby hexagonal site to a S vacancy were 0.89, 0.63, and 0.29 eV for Cu, Ag, and Au, respectively. These energies were needed to overcome and return to the initial states. According to the Arrhenius law, these results suggest that the diffusion of the metal atoms from the IS to FS occurred in the range of seconds at room temperature for Ag and Au, and above 350 K for Cu. Thus, it can be assumed that substitutional atoms would remain fixed to the vacancies at temperatures above 700 K.

WS_2_ monolayers with well-distributed S vacancies are therefore suitable substrates for anchoring single metal atoms. In addition, mono-vacancies (*V_S_*) and di-vacancies (*V_2S2_*) of S on WS_2_ formed new states in the energy gap of the monolayer (Figure 1). Band gaps of 1.20 and 1.0 eV were found for WS_2_-*V_S_* and WS_2_-*V_2S2_*, respectively (compared to 1.85 eV for pristine WS_2_, as shown in Figure 1), suggesting that they are still semiconductors. These peaks were obtained for W (*5d*), just above the Fermi energy. These empty *5d* states were mainly provided by the orbitals of the tungsten atoms closest to the vacancies. It was also observed that for the WS_2_-*V_2S2_*, there were empty *3p* states provided by the S atoms closest to the vacancies (see Figure 1). The crystal structures of the WS_2_-*V_S_* and WS_2_-*V_2S2_* did not change significantly. These results were consistent with the results of other research groups [13].

Similarly, coinage metal atoms *M_S_* placed into S mono-vacancies introduced a change in semiconductor n-type doping (see Figure 3). Figure 3 shows how the DOS moves to lower energy values. Simultaneously, the conduction band passed through the Fermi level, indicating that Cu, Ag, or Au doping comprises strong n-type doping. This phenomenon is expected to enhance the electrical conductivity of WS_2_. Zhang et al. demonstrated that Cu-doping could enhance the electrical conductivity of WS_2_ [43]. Based on a PDOS analysis, the *3d* and *4p* orbitals of Cu and the *5d* orbital of W exhibited substantial overlap in the vicinity of the Fermi level. This observation indicates a pronounced hybridization between these orbitals (as depicted in Figure 3a), a trend that was notably similar to the results observed for the Ag atom placed within the S vacancy (see Figure 3b).

The *6s* orbital of Au moves toward a *5d* orbital near the Fermi level, as shown in Figure 3c. This phenomenon is particularly pronounced in atoms characterized by a high effective nuclear charge, e.g., Cu, where the gap between the *3d* and *4s* orbitals is significantly larger than the corresponding gap between the *5d* and *6s* orbitals, as in Au (refer to Figure 3c). Subsequently, the O_2_, NO, and CO molecules interacting with the monolayers WS_2_-*M_S_* (M = group 11) were fully optimized, starting from different initial structures; however, only the ground states were studied in this work. We observed that the molecules were adsorbed on the preferential interaction region created by the substitutional defect, with a single state of adsorption on each metal atom (Figure 4). The adsorption energies of CO or NO molecules on Au, Ag, and Cu placed into a S vacancy, which agree with the chemisorption regime, were competitive (Table 1). As a consequence, the adsorption energies provided evidence that nitric oxide and carbon monoxide were more chemisorbed than O_2_ (Table 1). Therefore, it can be inferred that O_2_ adsorption cannot effectively compete with carbon monoxide or nitric oxide adsorption. These findings suggest that O_2_ would not act as inhibitor for NO and CO adsorption.

In contrast, our group recently demonstrated that the adsorption of CO and NO on MoS_2_-*Ag_S_* and MoS_2_-*Au_S_* falls within the physisorption regime [15], as compared to the materials currently under investigation. Thus, chemical activation of the WS_2_ monolayer was induced by substitutional metal atoms. The most appropriate adsorption mode of CO and NO showed that the C or N atom attached to a metal atom and with Metal-C (or N) bonded perpendicular to the monolayer. The shortest bond lengths between Cu, Au, and Ag atoms and the CO and NO molecules are 1.86 and 1.79, 2.0, 2.0, 2.1, and 2.1 Å, respectively (Figure 4). The bond lengths for carbon oxide and nitric oxide on WS_2_-*Cu_S_* are 1.14 and 1.18 Å, respectively. On WS_2_-*Au_S_*, these lengths are 1.15 and 1.18, while on WS_2_-*Ag_S_*, they are 1.14 and 1.19 Å, respectively. In comparison, in the gas phase, these bond lengths are typically measured as 1.12 Å for carbon oxide and 1.15 Å for nitric oxide, respectively.

Similarly, an adsorbed O_2_ molecule results in an elongated bond length relative to that obtained in the gas phase (1.23 Å). The 1.32, 1.30, and 1.33 Å bond lengths for O-O, calculated with WS_2_-*Ag_S_*, WS_2_-*Cu_S_*, and WS_2_-*Au_S_*, suggest the superoxide state of O_2_. It was discovered that the oxygen molecule did not react with the pre-adsorbed carbon monoxide or nitric oxide on WS_2_-*M_S_*, and neither nitric oxide nor carbon monoxide reacted with the adsorbed O_2_ molecule. Furthermore, the results revealed that both CO and NO molecules were adsorbed without dissociation (Figure 4). Consequently, from a statistical perspective, it is more plausible that the O_2_ molecule was initially desorbed before being displaced by a CO or NO molecule.

The observed enlargement in the O_2_, NO, and CO molecules was attributed to a charge transference occurring in the defective WS_2_ monolayer, donation, and back donation. This charge transference could be detected via the occupation of the defective states on the WS_2_, as well as via the electric transport properties, conductivity and/or resistivity. Figure 5k shows that the adsorption of CO molecule leads to modified electronic properties of WS_2_-*Au_S_* (see Figure 5i). The DOS for spin-up-down revealed *6s* (red color) and *6p* (colored in green color) bands of Au, i.e., above the Fermi level at the conduction band minimum (CBM), vanishing for WS_2_-*Au_S_* (Figure 5i,k). Likewise, the PDOS for W, S, *Au_S_*, C, and O atoms suggested C(*2p*)-Au(*6s,5d*)-W(*5d*) hybridization at the Fermi level and above it for spin-up and spin-down, respectively. Total magnetic moments were computed at 0.0 and 0.0 μ_B_ for WS_2_-*Au_S_* and WS_2_-*Au_S_*-CO, respectively (Table 1). This result indicated that the system with substitutional gold increased in resistivity due to CO adsorption. In contrast, the DOS and PDOS in Figure 5a,c for WS_2_-*Cu_S_* and WS_2_-*Cu_S_*-CO show that the adsorption of the CO molecule did not induce any modification in the electronic properties. Likewise, there were no changes in the electronic structure for WS_2_-*Ag_S_* or WS_2_-*Ag_S_*-CO, as shown in Figure 5e,g, respectively.

In contrast, when NO was adsorbed onto Cu, Au, and Ag, it introduced a new state in the band gap of the WS_2_ monolayer (Figure 5). Likewise, the PDOS for W, S, *Cu_S_*, N, and O atoms suggested O(*sp*)-N(*sp*)-Cu(*3d*)-W(*5d*) hybridization for the new state introduced below the Fermi level and O(*p*)-Cu(*3d*)-W(*5d*) hybridization for states above the Fermi energy, with spin-up and spin-down, respectively (Figure 5a,b). This outcome implies that the system with substitutional metal atoms decreased its resistivity by adsorbing NO molecules. The magnetic moments per cell were 0.0 and 1.93 μ_B_ for WS_2_-*Cu_S_* and WS_2_-*Cu_S_*-NO, respectively. The electronic behavior remained essentially consistent for the adsorption of NO on *Ag_S_* (Figure 5e,f). In particular, *Au_S_* introduced Au (*6s*) state contributions in its hybridization to bond with the *V_S_* vacancy (Figure 5i,j). The magnetic moments per cell were 0.0/0.0 and 1.92/1.82 μ_B_ for WS_2_-*Au_S_/Ag_S_* and WS_2_-*Au_S_*-NO/WS_2_-*Ag_S_*-NO, respectively.

Furthermore, in the case of O_2_ molecules adsorbed on *Cu_S_*, a transition from semiconductor n-type doping (see Figure 5a) to half-metal (Figure 5d) was induced, which was related to the hybridization of *3d* states of copper and *p* orbitals of oxygen atoms in WS_2_-*Cu_S_*-O_2_, respectively (Figure 5d*)*. The bandgap for spin-up was approximately 0.7 eV for MoS_2_-*Cu_S_*-O_2_ systems. A Lowdin charge analysis indicated that approximately 0.18 eV was transferred from the WS_2_-*Cu_S_* system to the O_2_ molecule. (refer to Table 1). Therefore, if the experimental system was exposed to air, the low quantities of O_2_ molecules present should have effected the oxidation of the Cu atoms. This observation may be linked to the potential gas sensing properties of this material. Conversely, when O_2_ was adsorbed onto *Ag_S_*/*Au_S_*, a new state in the band gap of WS_2_-*Ag_S_*/WS_2_-*Au_S_* monolayer (Figure 5) was introduced. This resulted in an excess of polarized electrons, as indicated by a total magnetization exceeding 1 μ_B_ (See Table 1).

Charge transference occurred from NO and CO adsorbed molecules to the defective WS_2_ monolayer (refer to Table 1). In contrast, there were electron transfers from the d orbital state of the metal atoms (Cu, Au, or Ag) to the 2π* anti-bonding orbital of O_2_ This behavior could be explained as follows: the calculated vertical electronic affinities for WS_2_-*Au_S_*, WS_2_-*Cu_S_*, and WS_2_-*Ag_S_* were −0.19, −0.14 and −0.160 eV, respectively. In contrast, the vertical ionization potentials for the same systems were calculated to be 1.88, 2.10, and 1.75 eV, respectively. In comparison, the electron affinity (ionization potential) values for O_2_, NO, and CO were 1.08 (12.63), 1.17 (9.92), and 2.68 (13.82) eV, respectively. Consequently, the possibility of removing electrons from the adsorbed species was hindered.

The adsorption energies for molecular oxygen, carbon monoxide, and nitric oxide on *Ag_2S2_*, *Cu_2S2_*, and *Au_2S2_* dimers positioned within two S vacancies are shown in Table 1 (see Figure 6). Additionally, it was found that pairs of molecules, i.e., O_2_-NO and O_2_-CO, could be adsorbed on *Au_2S2_*, *Cu_2S2_*, and *Ag_2S2_*. The shortest bond lengths between the *Au_2S2_* defect and the carbon monoxide and nitric oxide molecules are 2.21 (Au-C) and 2.09 (Au-N) Å, respectively (Figure 6g,h) or 2.28 (Ag-C) and 2.15 (Ag-N) Å, 1.84 (Cu-C) and 1.93 (Cu-N) Å, respectively (see Figure 6). According to the most likely adsorption mode of carbon monoxide, the C atom, bonded with the placed Au *Au_S_*, formed an Au-C-Au bridge. Figure 6e illustrates the same geometric structure for the adsorption of CO on *Ag_2S2_*. In contrast, when CO adsorbed on *Cu_2S2_*, the C atom bonded with the *Cu_S_* defect, positioning the molecule perpendicular to the monolayer (Figure 6b). For the nitric oxide molecule, it formed Au-N-O-Au and Ag-N-O-Ag bridges parallel to the monolayer. Additionally, this molecule formed a Cu-N-Cu bridge perpendicular to the monolayer. Furthermore, O_2_ formed Au-O-O-Au, Ag-O-O-Ag, and Cu-O-O-Cu bridges (see Figure 6). The shortest bond lengths between the *Ag_2S2_*, *Cu_2S2_*, and *Au_2S2_* dimer and O_2_ molecules were computed as 2.19, 1.89, and 2.14 Å, respectively.

Larger distances were computed for the carbon monoxide and nitric oxide molecules relative to those found in the gas phase, as shown by the interaction between these dimers and the placed *Au_2S2_* dimer. The interatomic distances for CO and NO on MoS_2_-*Au_2S2_* were calculated as 1.17 and 1.23 Å, respectively. In comparison, in the gas phase, these bond lengths were about 1.12 and 1.15 Å, respectively. This may be related to the chemical activation of the adsorbed molecules. Table 2 shows the interatomic bond lengths for all cases.

Table 2 shows that O_2_ molecules were chemisorbed more strongly than CO or NO on *Au_2S2_*, *Cu_2S2_*, or *Ag_2S2_* placed into two S vacancies. Thus, if the experimental system was exposed to air, the low quantities of O_2_ molecules present likely effected the oxidation of the metallic atoms. According to the Arrhenius law, the results suggest that the CO, NO, and O_2_ desorption (equal to the adsorption energy) occurred at temperatures between 400–500 K. The proposed approach could therefore be used to develop sensor devices based on a specific temperature differential.

Figure 7 shows the DOS and PDOS densities of states for the defective systems. In the case of WS_2_-*M_2S2_*-CO systems (M = Cu, Ag, and Au), the DOS peaks for spin-up and spin-down at the conduction band minimum were very close, so the electronic changes were almost negligible (See Figure 7c,g,k). Additionally, the PDOS for W, S, *Au_S_*, C, and O atoms evidenced O(*p*)-C(*s,p*)-Au(*6s,5d*)-W(*5d*) hybridization at the Fermi level and above it for spin-up and spin-down, respectively (Figure 7k). The magnetic moments per cell were found to be 0.00 and 0.29 μ_B_ for WS_2_-*Au_2S2_* and WS_2_-*Au_2S2_*-CO, respectively. The WS_2_-*Cu_2S2_*-CO system displayed similar density-of-states behavior to the WS_2_-*Au_2S2_*-CO system (Figure 7c). For the WS_2_-*Ag_2S2_* and WS_2_-*Ag_2S2_*-CO systems, no changes were observed in the DOS or PDOS at the Fermi level (Figure 7g). Conversely, when NO was adsorbed on substitutional *Au_2S2_*, it introduced a new level within the band gap of WS_2_-*M_2S2_* (M = Cu, Au, and Ag), resulting in the generation of spin polarization due to the presence of unpaired electrons (refer to Figure 7b,f,j). This effect led to a total magnetic moment of approximately 1.27, 1.75, and 1.84 μ_B_ for the WS_2_-*Cu_2S2_-*NO, WS_2_-*Ag_2S2_-*NO, and WS_2_-*Au_2S2_-*NO systems. The peak of spin-up at the Fermi level vanished, whereas a new state was shown at the valence band maximum of the MoS_2_-*M_2S2_*-NO (M = group 11) system. Our investigation into the interaction of these systems with oxygen molecules holds significant technological relevance, as even a trace amount of oxygen in the gas phase can potentially trigger metal oxidation. The O_2_ molecule, adsorbed on *Au_2S2_* or *Cu_2S2_* placed into S vacancies, induced a transition to half-metal for spin-up electrons, related to the hybridization of Au(*s*,*d*) or Cu(*s*,*d*) and O(*s*,*p*) orbitals, respectively (Figure 7d,l).

The bandgaps for WS_2_-*Au_2S2_*-O_2_ and WS_2_-*Cu_2S2_*-O_2_ systems are 0.7 and 0.6 eV, respectively. They also transform into magnetic systems, as shown in Table 2. O_2_ adsorbed on substitutional *Au_2S2_*, *Cu_2S2_* created a new level in the band gap of the WS_2_-*M_2S2_* (M = Cu and Au) monolayers (Figure 7d,l), which induced spin polarization. This phenomenon is similar to the one reported in the case of modified MoS_2_ with coinage atoms placed into S vacancies [15]. Similarly, Li et al. also reported that V, Nb, and Ta doped WS_2_ systems exhibited half-metal properties [44].

From all the above, the formation of substitutional defects of coinage metal atoms (Cu, Ag, and Au) placed in S mono- and di-vacancies were used to activate the monolayer chemically. The binding energies suggest that several atoms anchored on the WS_2_ monolayers were stable at 700 K. Additionally, the adsorption energies showed that nitric oxide and carbon oxide molecules exhibited stronger chemisorption than O_2_ molecules on any of the metal atoms placed in the S vacancies of the WS_2_ monolayer. The adsorption energies suggest that the NO, CO, and O_2_ molecules could be absorbed at room and high temperatures. The findings reported in the current study provide helpful information for sensing devices and the characterization of defective WS_2_ under different experimental conditions, i.e., when exposed to atmospheres containing O_2_ or CO.

## 3. Theoretical Method

Spin-polarized density functional theory method PBE-D3(BJ)/PAW, with correction due to dispersion interactions, was used as implemented in Quantum Espresso 6.1 [45]. The nonlinear exchange-correlation correction was included for W, S, Au, Cu, and Ag to improve the quality of core-valence interactions. The pseudopotentials obtained through the projector augmented wave (PAW) method were used with the following valence electronic states: W: *5s^2^5p^6^6s^2^6p^0^5d^4^*, S: *3s^2^3p^4^3d^0^*, Au: *6s^1^6p^0^5d^10^*, Ag: *5s^1^5p^0^4d^10^*, Cu: *4s^1^4p^0^3d^10^*, O: *2s^2^2p^4^*, C: *2s^2^2p^2^*, and N: *2s^2^2p^3^*. The 4 × 4 × 1 supercell was found to be big enough to hold the molecules under study (O_2_, NO, and CO). For energy convergence, a convergence threshold of 1.0 × 10−7 a.u. was assumed, as well as an energy cut-off of 60 Ry. The Broyden–Fletcher–Goldfarb–Shanno quasi-Newton algorithm was used for structural optimization. In the case of structural optimization, thresholds of 1.0 × 10^−5^ a.u. and 1.0 × 10^−4^ a.u. for total energy and residual forces were considered. A × 5 × 1 k-point grid, set by the Methfessel-Paxton method, was assumed. The grid was increased to 16 × 16 × 1 for PDOS and DOS calculations.

On the other hand, a defect *X_y_* within the Kröger–Vink notation refers to X species in the current case, i.e., metal atoms M_n_ (n = 1, 2) or vacancies V_n_ (n = 1, 2) occupying the site of the Y species in WS_2_ ML S atoms S_n_ (n = 1, 2) in the current work. Multiple S vacancies are denoted as *V_Sn_* (n = 1, 2), whereas placed clusters occupying S vacancies were considered *M_nSn_* (n = 1, 2, M = group 11). The adsorption energy of a metal atom or cluster placed into mono- or di-sulfur vacancies, as well as adsorption energies of the diatomic molecules on metal, were computed as follows:E_Metal-ads_ = E(WS_2_-*V_nSn_*) + E(M_n_) − E(WS_2_-*M_nSn_*)(1)
E_AB-ads_ = E(WS_2_-*M_nSn_*) + E(AB) − E(WS_2_-*M_nSn_*–AB)(2)
where E(WS_2_-*V_nSn_* + AB), E(WS_2_-*M_nSn_*), and E(WS_2_-*M_nSn_* − AB) represent the electronic energies of modified monolayers, i.e., S vacancies *V_nSn_* with embedded clusters *M_nSn_* (n = 1, 2, M = group 11) occupying S vacancies and interacting with AB diatomic molecules, respectively. Structures with two embedded metal atoms *M_nSn_* (n = 1, 2) were also studied. Additionally, Lowdin charges were computed for further analysis.

## 4. Conclusions

Modified monolayers of tungsten disulfide were studied using dispersion-corrected density functional theory, both isolated and interacting with molecules in the gas phase. These were considered as substitutional defects of coinage metal atoms (Cu, Ag and Au) placed in S mono- and di-vacancies, leading to WS_2_-*M_nSn_* (n = 1, 2; M = Cu and Au) systems. Binding energies, as well as diffusion energy barriers computed for WS_2_-*M*_S_ (M = Cu and Au), point to the stability of the atoms in S vacancies, even at temperatures above 700 K.

These stable defects were investigated as the preferred regions of interaction with CO, NO, and O_2_ molecules in the gas phase. NO and CO molecules exhibited stronger chemisorption on WS_2_-*M_nSn_* (n = 1, 2; M = Cu and Au) systems than O_2_. Furthermore, based on a simultaneous adsorption simulation, O_2_ did not displace either NO or CO molecules. The DOS and PDOS revealed that the modified tungsten disulfide monolayers were highly sensitive to the molecule adsorbed on the metal defect. The interaction of NO with the defective monolayers introduced defect states and total magnetic moments.

Therefore, these materials are proposed as promising candidates for electronic or magnetic devices in sensing applications, particularly for detecting molecules like NO. In particular, CO introduced changes in the electronic structure of MoS_2_-*Au_S_* as another potential sensing material. In addition, the adsorption of O_2_ molecules onto WS_2_-*Au_2S2_* and WS_2_-*Cu_S_* introduced a half-metallic behavior, which holds significance for spintronics applications. This is because the introduced defect states alone enhanced the conduction of spin-up electrons. This study sheds light on the intricate interplay between modified tungsten disulfide monolayers and various gas-phase molecules. The investigation of substitutional defects involving coinage metal atoms within S vacancies has revealed the potential of these systems in the development enhanced sensing applications, particularly in the realm of NO detection. The observed chemisorption behavior, electronic structure changes, and induced magnetic moments provide valuable insights into the promising role of these materials in the design of electronic, magnetic, and spintronic devices. The findings presented here contribute to our understanding of the interactions between defects and adsorbed molecules and open doors for further exploration and innovation in the field of nanoscale functional materials.

## Figures and Tables

**Figure 1 ijms-24-15079-f001:**
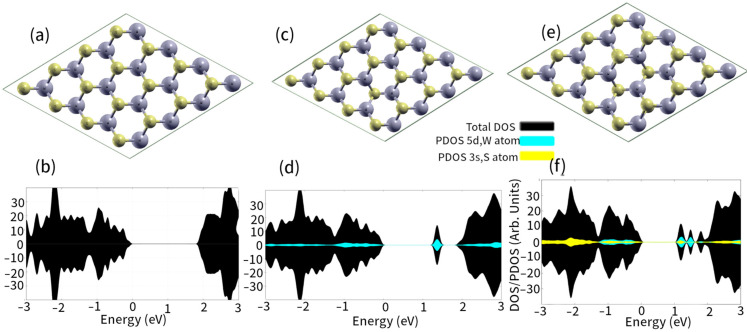
Top views of the fully optimized WS_2_ (**a**), WS_2_-*V_S_* (**c**), and WS_2_-*V_S_* (**e**) supercells at the PBE-D3/PAW level of theory. DOS and PDOS were calculated for WS_2_ (**b**), WS_2_-*V_S_* (**d**), and WS_2_-*V_S_* (**f**) supercells. Fermi energy set at 0 eV. PDOS projected over *3s* and *5d* orbitals of S and W atoms, respectively.

**Figure 2 ijms-24-15079-f002:**
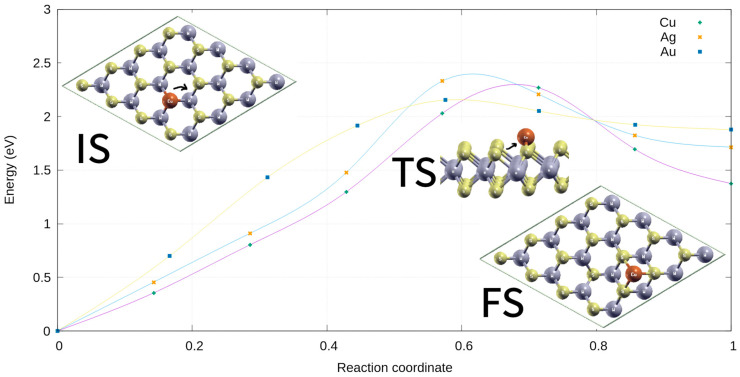
Schematic representation of the minimum energy pathway (MEP) for diffusion energy barriers from S vacancies to a nearby hexagonal site for Cu, Ag, and Au, respectively. The top view configurations corresponding to the states are also shown. Only WS_2_-*M* (M = Cu and Au) systems were considered.

**Figure 3 ijms-24-15079-f003:**
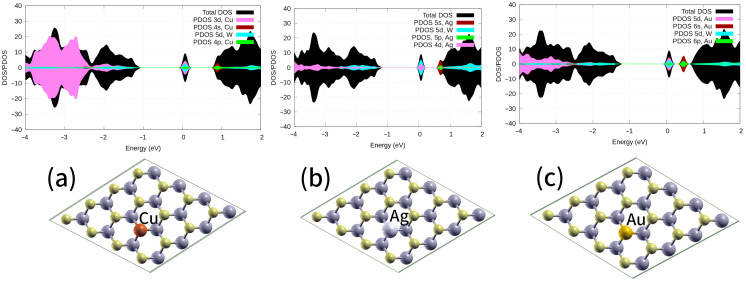
DOS and PDOS, in arbitrary units, for WS_2_ monolayers with defect (**a**) *Cu_S_*, (**b**) *Ag_S_*, and (**c**) *Au_S_* are shown. PDOS projected over *5d* of W *s*, *p*, and *d* orbitals of Cu, Ag, and Au atoms, respectively. Fermi energy set at 0 eV. Values were computed using the PBE-D3 level of theory. Insets: top views of the fully optimized defective systems.

**Figure 4 ijms-24-15079-f004:**
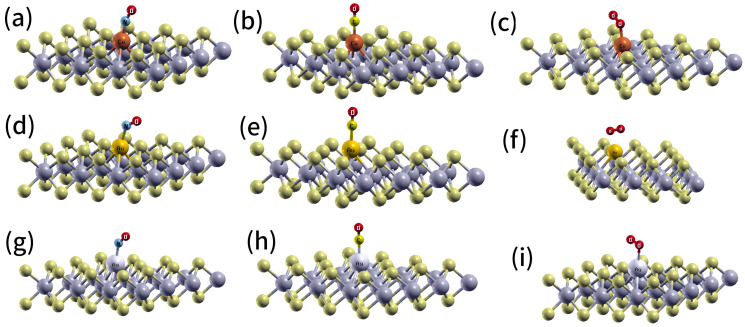
Structures obtained for the modified WS_2_-*M_S_* monolayers (M = Cu, Au, and Ag) interacting with CO, NO, and O_2_. (**a**) WS_2_-*Cu_S_*-NO, (**b**) WS_2_-*Cu_S_*-CO, (**c**) WS_2_-*Cu_S_*-O_2_, (**d**) WS_2_-*Au_S_*-NO, (**e**) WS_2_-*Au_S_*-CO, (**f**) WS_2_-*Au_S_*-O_2_, (**g**) WS_2_-*Ag_S_*-NO, (**h**) WS_2_-*Ag_S_*-CO, (**i**) WS_2_-*Ag_S_*-O_2_.

**Figure 5 ijms-24-15079-f005:**
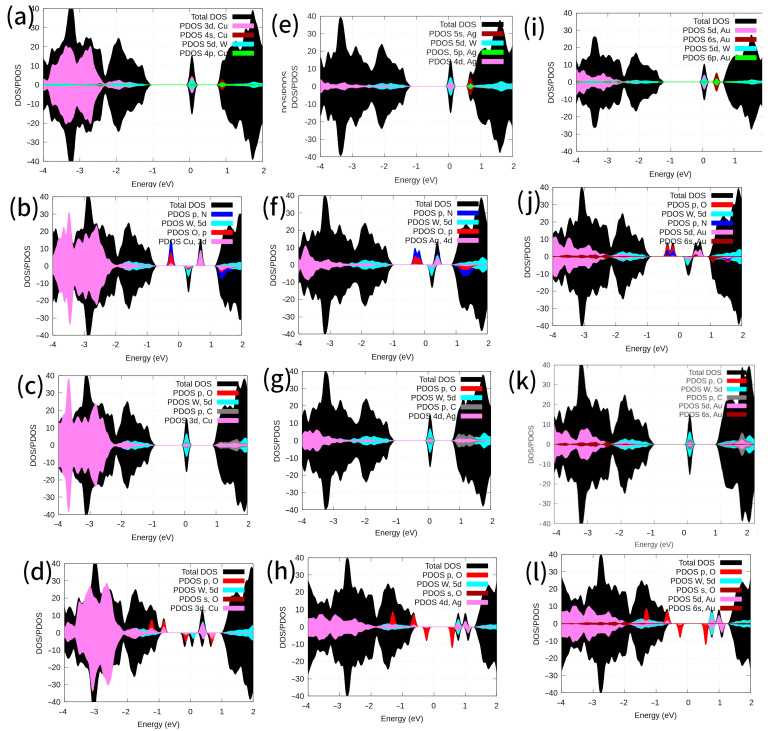
DOS and PDOS for the modified systems are shown. PDOS is projected onto the annotated orbitals. The Fermi energy is set at 0 eV. (**a**) WS_2_-*Cu_S_*, (**b**) WS_2_-*Cu_S_*-NO, (**c**) WS_2_-*Cu_S_*-CO, (**d**) WS_2_-*Cu_S_*-O_2_, (**e**) WS_2_-*Ag_S_*, (**f**) WS_2_-*Ag_S_*-NO, (**g**) WS_2_-*Ag_S_*-CO, (**h**) WS_2_-*Ag_S_*-O_2_, (**i**) WS_2_-*Au_S_*, (**j**) WS_2_-*Au_S_*-NO, (**k**) WS_2_-*Au_S_*-CO, (**l**) WS_2_-*Au_S_*-O_2_.

**Figure 6 ijms-24-15079-f006:**
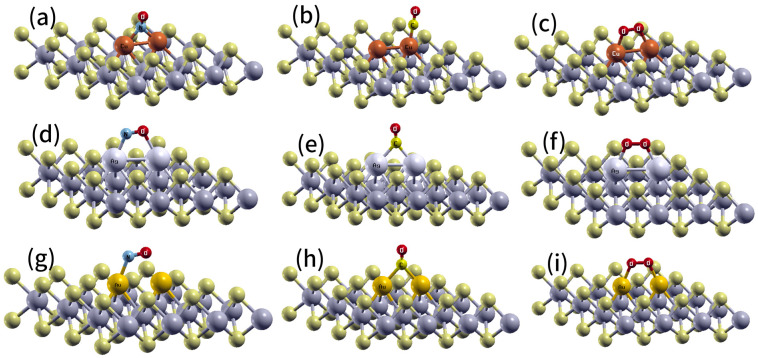
Structures obtained for the defective WS_2_-*M_2S2_* (M = Cu, Ag, and Au) monolayers interacting with CO, NO, and O_2_. (**a**) WS_2_-*Cu_2S2_*-NO, (**b**) WS_2_-*Cu_2S2_*-CO, (**c**) WS_2_-*Cu_2S2_*-O_2_, (**d**) WS_2_-*Ag_2S2_*-NO, (**e**) WS_2_-*Ag_2S2_*-CO, (**f**) WS_2_-*Ag_2S2_*-O_2_, (**g**) WS_2_-*Au_2S2_*-NO, (**h**) WS_2_-*Au_2S2_*-CO, (**i**) WS_2_-*Au_2S2_*-O_2_.

**Figure 7 ijms-24-15079-f007:**
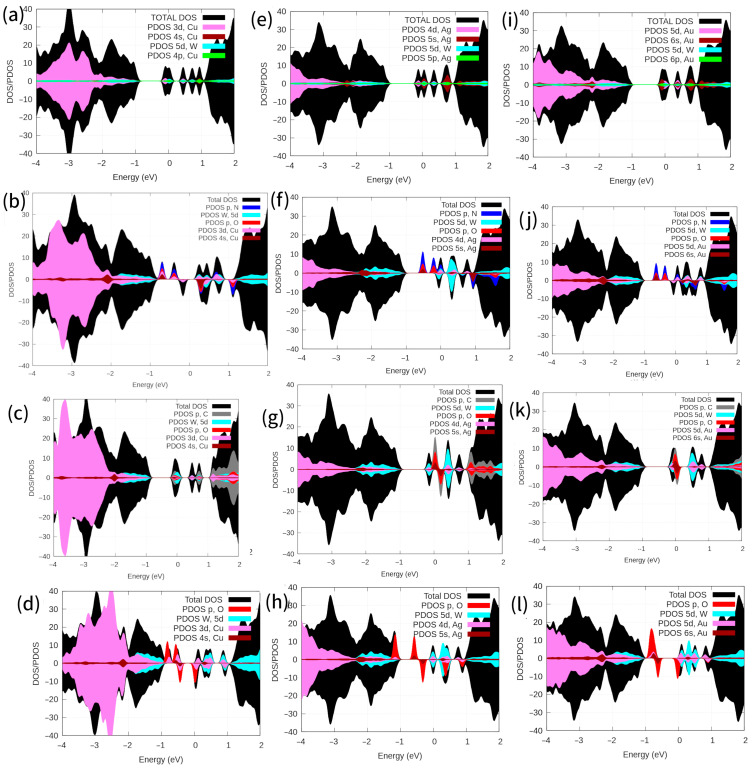
DOS and PDOS for the defective systems are shown. PDOS is projected onto the annotated orbitals. The Fermi energy was set at 0 eV. (**a**) WS_2_-*Cu_2S2_* (**b**) WS_2_-*Cu_2S2_*-NO (**c**) WS_2_-*Cu_2S2_*-CO, (**d**) WS_2_-*Cu_2S2_*-O_2_, (**e**) WS_2_-*Ag_2S2_* (**f**) WS_2_-*Ag_2S2_*-NO (**g**) WS_2_-*Ag_2S2_*-CO, (**h**) WS_2_-*Ag_2S2_*-O_2_, and (**i**) WS_2_-*Au_2S2_* (**j**) WS_2_-*Au_2S2_*-NO (**k**) WS_2_-*Au_2S2_*-CO, (**l**) WS_2_-*Au_2S2_*-O_2_.

**Table 1 ijms-24-15079-t001:** Adsorption energy, in eV, of diatomic molecules on modified WS_2_-*M_S_* monolayers with a single substitutional atom.

System	E_ads_ (eV)	Total Magnetization (μ_B_)	Total Charge on AB Molecule (e)	A–B Bond Lengths (Å)
WS_2_-*Cu_s_-*O_2_	1.08	1.03	−0.18	1.30
WS_2_-*Cu_s_-*CO	1.38	0.0	0.307	1.15
WS_2_-*Cu_s_-*NO	1.56	1.93	0.07	1.18
WS_2_-*Ag_s_-*O_2_	0.85	1.05	−0.22	1.32
WS_2_-*Ag_s_-*CO	0.91	0.0	0.34	1.14
WS_2_-*Ag_s_-*NO	1.02	1.82	0.05	1.19
WS_2_-*Au_s_-*O_2_	0.94	1.04	−0.22	1.33
WS_2_-*Au_s_-*CO	1.19	0.0	0.28	1.15
WS_2_-*Au_s_-*NO	1.26	1.92	0.05	1.18

**Table 2 ijms-24-15079-t002:** Adsorption energy in eV of diatomic molecules on defective WS_2_-*M_2S2_* MLs with two substitutional atoms, total magnetization (μ_B_), total charge on AB molecule, and bond lengths in Å.

System	E_ads_ (eV)	Total Magnetization (μ_B_)	Total Charge on AB (e)	A–B Bond Lengths Å
MoS_2_-*Cu_2S2_-*O_2_	2.04	0.63	−0.42	1.38
MoS_2_-*Cu_2S2_-*CO	1.43	0.0	0.29	1.15
MoS_2_-*Cu_2S2_-*NO	1.90	1.27	−0.09	1.22
MoS_2_-*Ag_2S2_-*O_2_	1.42	0.53	−0.33	1.34
MoS_2_-*Ag_2S2_-*CO	0.93	0.15	0.21	1.16
MoS_2_-*Ag_2S2_-*NO	1.08	1.75	−0.09	1.23
MoS_2_-*Au_2S2_-*O_2_	1.36	0.97	−0.41	1.38
MoS_2_-*Au_2S2_-*CO	0.77	0.29	0.11	1.17
TMoS_2_-*Au_2S2_-*NO	1.12	1.84	−0.08	1.23

## Data Availability

The data presented in this study are available on request from the corresponding author.

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
