# Peer review of "Gas Sensing and Half-Metallic Materials Design Using Metal Embedded into S Vacancies in WS2 Monolayers: Adsorption of NO, CO, and O2 Molecules"

_ijms, 2023, doi:10.3390/ijms242015079_

Round 1

Reviewer 1 Report

This study investigates the adsorption of CO, NO, and O2 molecules on Cu, Ag, and Au atoms embedded in S vacancies of a WS2 monolayer using dispersion-corrected density functional theory. The results reveal strong chemisorption of NO and CO molecules compared to O2, indicating potential applications in designing sensing devices for atmospheric pollutants and suggesting potential for spintronics applications when O2 is adsorbed on specific configurations. This manuscript is comprehensive and interesting to a wide range of researchers. I have a few questions.

1. In addition to the significance in the gas sensing field, the adsorption study of metal on TMDs is also important to the contact interface study for electronic devices. I suggest the authors add another paragraph in the introduction to discuss the significance of this work to the TMD electronic device field. Some references need to be discussed: 

a. Chen, J.; Zhang, Z.; Guo, Y.; Robertson, J. Metal Contacts with Moire Interfaces on WSe2 for Ambipolar Applications. Appl. Phys. Lett. 2022, 121, 051602.

b. Wang, X.; Kim, S. Y.; Wallace, R. M. Interface Chemistry and Band Alignment Study of Ni and Ag Contacts on MoS2. ACS Appl. Mater. Interfaces 2021, 13, 15802–15810.

c. Wang, Y.; Chhowalla, M. Making Clean Electrical Contacts on 2d Transition Metal Dichalcogenides. Nat. Rev. Phys. 2022, 4, 101-112.

d. Schuler, B.; Qiu, D. Y.; Refaely-Abramson, S.; Kastl, C.; Chen, C. T.; Barja, S.; Koch, R. J.; Ogletree, D. F.; Aloni, S.; Schwartzberg, A. M.; Neaton, J. B.; Louie, S. G.; Weber-Bargioni, A. Large Spin-Orbit Splitting of Deep in-Gap Defect States of Engineered Sulfur Vacancies in Monolayer WS2. Phys. Rev. Lett. 2019, 123

2. The authors have done extensive calculations and presented profound results. However, the Figures are not very reader friendly. The font size is very small in Figures 3. The resolution of Figure 4, 6, and 7 needs to be improved.

3. The authors need to develop a stronger connection between the calculation results (DOS, adsorption energy, etc.) and the gas sensing properties.

4. Has it been studied that the adsorption properties of gases on defect-free WS2 in the literature?

Author Response

Response Letter.

Below are listed the answers to the reviewer's comments. Changes in the manuscript are highlighted in green.

Response to Reviewer #2:

  1. In addition to the significance in the gas sensing field, the adsorption study of metal on TMDs is also important to the contact interface study for electronic devices. I suggest the authors add another paragraph in the introduction to discuss the significance of this work to the TMD electronic device field. Some references need to be discussed:

Author’s reply. We agree with the reviewer’s suggestion. Consequently, the following paragraph was added. Also, additional references were included. References in this letter are annotated in APA format only for simplicity.

  • Since the transition metal dichalcogenides (TMDCs) have been proposed to be part of electronic devices (Bhattacharyya & Acharyya, 2021; Cho et al., 2015; H. Li et al., 2012; Sarkar et al., 2014; Wu et al., 2022), with conductance near the quantum limit (Y. Wang & Chhowalla, 2021), the analysis of ultra clean contacts as well as surface defects has been focused by the scientific community (Chen et al., 2022). Since one of the most challenging problems to face by TMDCs-devices is the high contact resistance, several metals have been tested (Chen et al., 2022; Yang et al., 2023). Matching alignments, as well as contact metals could lead to chemisorption or physisorption. For instance, Wang and coworkers determined that Ag leads to Ohmic contacts with MoS2 bulk, as well as forms van der Waals interactions in the interface with MoS2 bilayers (X. Wang et al., 2021). Also, Ohmic contacts have been reported for WS2/Zr2C and WS2/Hf3C2 systems (C. Li et al., 2020). Similarly, ab initio calculations were performed on WSe2 to determine (Yang et al., 2023)

  • The changes introduced by sulfur vacancies on WS2 monolayers have been studied, even by experimental approaches. For instance, Schuler reported micrographs of CVD-grown monolayers of WS2 obtained by means of CO-tip non contact atomic force microscopy and scanning tunneling microscopy/spectroscopy (STS). Results obtained by STS revealed a characteristic fingerprint caused by an extraordinarily strong spin-orbit coupling, confirmed by DFT and GW calculations (Schuler et al., 2019). Similar behaviors have been identified in case of Mn-doped monolayers of WS2, in which impurity d states appeared within the bandgap. Magnetic properties were introduced as well (Zhao et al., 2015).

  1. The authors have done extensive calculations and presented profound results. However, the Figures are not very reader friendly. The font size is very small in Figures 3. The resolution of Figure 4, 6, and 7 needs to be improved.

Authors reply. All the figures were improved as required. Color key in Figure 3 was replaced.

Figure 3.

  1. The authors need to develop a stronger connection between the calculation results (DOS, adsorption energy, etc.) and the gas sensing properties.

Author’s reply. The following paragraphs were initially included and highlighted phrases were added to point to the gas sensing properties suggested by our results.

On page 6 paragraph 1, it is mentioned:

The enlargement observed in the O2, NO and CO molecules was attributed to charge transference happening in the defective WS2 monolayer, donation, and back donation. This charge transference could be detected via the occupation of the defective states on the WS2, also electric transport properties, conductivity and/or resistivity. Likewise, PDOS for W, S, AuS, C, and O atoms suggest C(2p)-Au(6s,5d)-W(5d) hybridization at the Fermi level and above it for spin-up and spin-down, respectively.

The next sentence was added:

Likewise, PDOS for W, S, AuS, C, and O atoms suggest C(2p)-Au(6s,5d)-W(5d)  hybridization at the Fermi level and above it for spin-up and spin-down, respectively. This result suggests that the system with substitutional gold increases its resistivity by adsorbing the CO molecule.

On page 7 paragraph 1, it is mentioned:

Contrastingly, NO adsorbed-on Cu, Au and Ag introduces a new state in the band gap of the WS2 monolayer (Figure 5). Likewise, PDOS for W, S, CuS, N, and O atoms suggest O(sp)-N(sp)-Cu(3d)-W(5d) hybridization for the new state introduced below the Fermi level, and O(p)-Cu(3d)-W(5d) hybridization for states above the Fermi energy, with spin-up and spin-down, respectively.

The next sentence is added:

Contrastingly, NO adsorbed-on Cu, Au and Ag introduces a new state in the band gap of the WS2 monolayer (Figure 5). Likewise, PDOS for W, S, CuS, N, and O atoms suggest O(sp)-N(sp)-Cu(3d)-W(5d) hybridization for the new state introduced below the Fermi level, and O(p)-Cu(3d)-W(5d) hybridization for states above the Fermi energy, with spin-up and spin-down, respectively. This result suggests that the system with substitutional metal atom decreases its resistivity by adsorbing the NO molecule.

On page 7 paragraph 2, it is mentioned:

More interestingly, for O2 molecule adsorbed on CuS placed, induces a transition from semiconductor n-type doping (see Figure 5a) to half-metal (Figure 5d), which is related to hybridization of 3d states of copper and p orbitals of oxygen atoms in WS2-CuS-O2, respectively (Figure 5d). The bandgap for spin-up is approximately of about 0.7 eV for MoS2-CuS-O2 systems. The above can be correlated with potential gas sensing properties of this material.

  1. Has it been studied that the adsorption properties of gases on defect-free WS2 in the literature?

Author’s reply. Since the adsorption of some gases on defect-free WS2 monolayers have been reported, the following paragraph was added in page 2.

Also, gas sensing by two-dimensional materials exploits charge transfer processes that modify the resistance after exposure to gas molecules. For instance, in a previous report, density functional theory has been used to investigate the adsorption of CO, NH3, NO, and NO2 molecules on monolayer, bilayer, and heterobilayer of MoS2 and WS2 [9]. The calculated adsorption energies show that in each case the gas molecule is physisorbed , with values ranging from 90 to 200 meV. According to the Arrhenius law, these energies suggest that the molecules could not be absorbed at room temperature. In addition, has been reported the adsorption of CO, NH3 , NO, and NOmolecules on defects can modulate the magnetic and electronic properties of WS2 (Cui et al., 2023). However, the calculated adsorption energies suggest that the molecules could not be absorbed at room temperature, except for NO adsorbed-on VW and VS defects with 0.88 and 0.31 eV, respectively.

Reviewer 2 Report

The authors proposed that Cu, Ag and Au-modified WS2 monolayers are promising candidates as gas sensors for detecting toxic gas molecules like CO and NO. The findings elucidate an idea about the interaction between defects and adsorbed molecules and are explained by the DFT study. The proposed method is attractive regarding electronic, magnetic, and optical behaviors for the gas sensing application. This work can be considered for publication in this journal once the following issues have been addressed.

1. As the authors claimed, the main achievement of this report is to detect toxic gases by fabricated WS2 gas sensors in terms of the materials' electronic, magnetic, and optical behaviors. However, when compared to other reports, this finding is not new (Solid State Communications 2015, 215, 1-4). The authors should clearly highlight their critical findings and the advantages of their system.

2. The author posits that gas adsorption transfers charge from the WS2 monolayer to the adsorbed gas molecule. However, it's important to recognize that such computational results can be more intricate than the actual operational environment of gas sensors. The author should briefly clarify how vacancy defects could potentially enhance adsorption strength, stability, and overall sensing performance in practical applications.

Minor editing of English language required.

Author Response

(The authors gave the same response as above.)
